# Imbalanced data robust online continual learning based on evolving class aware memory selection and built-in contrastive representation learning

## Abstract

Continual Learning (CL) aims to learn and adapt continuously to new information while retaining previously acquired knowledge. Most state of the art CL methods currently emphasize class incremental learning. In this approach, class data is introduced and processed only once within a defined task boundary. However, these methods often struggle in dynamic environments, especially when dealing with imbalanced data, shifting classes, and evolving domains. Such challenges arise from changes in correlations and diversities, necessitating ongoing adjustments to previously established class and data representations. In this paper, we introduce a novel online CL algorithm, dubbed as Memory Selection with Contrastive Learning (MSCL), based on evolving intra-class diversity and inter-class boundary aware memory selection and contrastive data representation learning. Specifically, we propose a memory selection method called Feature-Distance Based Sample Selection (FDBS), which evaluates the distance between new data and the memory set to assess the representability of new data to keep the memory aware of evolving inter-class similarities and intra-class diversity of the previously seen data. Moreover, as the data stream unfolds with new class and/or domain data and requires data representation adaptation, we introduce a novel built-in contrastive learning loss (IWL) that seamlessly leverages the importance weights computed during the memory selection process, and encourages instances of the same class to be brought closer together while pushing instances of different classes apart. We tested our method on various datasets such as MNIST, Cifar-100, PACS, DomainNet, and mini-ImageNet using different architectures. In balanced data scenarios, our approach either matches or outperforms leading memory-based CL techniques. However, it significantly excels in challenging settings like imbalanced class, domain, or class-domain CL. Additionally, our experiments demonstrate that integrating our proposed FDBS and IWL techniques enhances the performance of existing rehearsal-based CL methods with significant margins both in balanced and imbalanced scenarios.

## 1 Introduction

Continual Learning (CL) assumes that a model learns from a continuous stream of data over time, without access to previously seen data. It faces the challenge of *catastrophic forgetting*, which occurs when a model forgets previously learned knowledge as it learns new information. State of the art has featured three major CL approaches (*e.g.*, Regularisation-based Kirkpatrick et al. (2017); Zenke et al. (2017); Chaudhry et al. (2018), Parameter isolation oriented Rusu et al. (2016); Verma et al. (2021); Singh et al. (2021)) and rehearsal-based Rolnick et al. (2019); Aljundi et al. (2019a;b); Yoon et al. (2022), along with various CL paradigms van de Ven & Tolias (2019) (*e.g.*, Task-incremental learning (TIL), Domain-incremental learning (DIL), and Class-incremental learning (CIL)). Early CL methods, *e.g.*, Kirkpatrick et al. (2017); Serrà et al. (2018), primarily adopted a task-incremental learning (TIL) paradigm and made the unrealistic assumption of having access to task boundaries not only during training for knowledge consolidation but also during inference. As a result, most recent research on CL has focused on class incremental learning (CIL), *e.g.*,Rebuffi et al. (2017); Gao

et al. (2023); Douillard et al. (2020); Lange & Tuytelaars (2021), which require the model to learn from a sequence of mutually class exclusive tasks and perform the inference without task boundary information. However, in such a scenario, each class can be learned only once within a task with all the class data assumed available for learning and thereby prevents further class adaptation when data distribution shifts for already seen classes come to occur, in particular with new domains. Furthermore, a vast majority of these CIL methods only consider balanced distribution over classes and tasks and are benchmarked using some single domain datasets, *e.g.*, Cifar, ImageNet, although streamed data distributions in CL are generally non-stationary in the real world. As a result, they face significant challenges in presence of imbalanced data in class and domain Wu et al. (2019)Liu et al. (2022). Ye et al. (2022) introduce a novel approach for quantifying dataset distribution shifts across two distinct dimensions. Their analysis highlights that datasets such as ImageNetVinyals et al. (2016) and CifarKrizhevsky (2009) primarily showcase correlation shifts, characterized by alterations in the relationship between features and labels. In contrast, datasets like PACSLi et al. (2017) and DomainNetPeng et al. (2019) predominantly exemplify diversity shifts, marked by the emergence of new features during testing.

In contrast to aforementioned CL methods, we consider in this paper a more general CL setting, namely task-free online CL (OCL), where data are streamed online through successive batches Aljundi et al. (2018); Zeno et al. (2021). They don't contain information about task boundaries and can be typically non-stationary as in real-life applications, thereby resulting in imbalanced data both in terms of classes and domains. Under such a setting, an ongoing batch of data can have none or much fewer samples for some classes than others. Furthermore, samples in a batch generally are not equally distributed over domains. As a result, seen classes can display more diversity and their boundaries can overlap and require to be refined, in particular when new domain and/or class data come to occur in the stream, thereby requiring dynamic adaptation of class and data representations.

Previous research (*e.g.*,Rolnick et al. (2019); van de Ven & Tolias (2019); Chrysakis & Moens (2020); Aljundi et al. (2019b)), has shown that rehearsal-based methods are more effective in mitigating catastrophic forgetting in various continual learning (CL) scenarios than other CL approaches. These methods utilize a small memory set to store and replay selected past data samples during current learning, enhancing the preservation of previously acquired knowledge. Consequently, the quality and composition of the samples stored in the memory set significantly influence the efficacy of rehearsal-based (CL) methods, especially in scenarios where data streams are non-stationary and exhibit imbalanced characteristics in terms of class and domain. However, most state of the art rehearsal-based CL methods only make use of very simple strategies to populate the memory set, ranging from random selection using Reservoir Sampling Rolnick et al. (2019) to herding-based approach Rebuffi et al. (2017) in selecting samples most similar to class prototypes within task boundaries. They are unaware of imbalanced data distributions and ignore increasing intra-class diversity and decreasing inter-class boundaries when new domain and/or class data occur over the course of incoming data streams as illustrated in Fig. 1 (a), thereby failing to adapt the previously acquired knowledge to novel data streams which require evolution of learned class boundaries.

In this paper, we argue that not all streamed data samples are equally beneficial for preserving and enhancing prior knowledge. The most valuable samples often capture the evolving diversity within classes and similarities between them. To harness this, we introduce a novel memory-based online CL approach, MSCL. This method has two core features: 1) **Dynamic Memory Population**: MSCL selects samples from incoming data streams that best represent diversity within classes and similarities between different classes. To achieve this, we've devised the Feature-Distance Based Sample Selection (**FDBS**). FDBS calculates an importance weight for each new sample based on its representational significance compared to the memory set in the feature space. Especially in imbalanced datasets, our method emphasizes diverse samples within each class and similar samples across different classes, ensuring a comprehensive memory set. 2) **Enhanced Data Representation with Contrastive Learning**: We've integrated a new Contrastive Learning Loss, **IWL**. This loss uses the importance weight from FDBS to bring similar class instances closer while distancing different class instances. By doing so, IWL refines class and data representations, boosting the efficacy of our CL approach. In essence, MSCL continually curates a memory set that captures the dynamic nature of data streams and refines data representation for optimal learning.

Our contributions are threefold:

- We design benchmarks for the problem of task free online CL with respect to imbalanced data both in terms of classes and domains, and highlight the limitations of existing CL methods in handling such complex non-stationary data.

- We introduce a novel replay-based online CL method, namely **MSCL**, based on: 1) a novel memory selection strategy, **FDBS**, that dynamically populates the memory set in taking into account intra-class diversity and inter-class boundary in the feature space, and 2) a novel data importance weight-based Contrastive Learning Loss, **IWL**, to continuously enhance discriminative data representation over the course of data streams.

- The proposed online CL method, **MSCL**, has been rigorously tested on a range of datasets through different architectures, and demonstrates its superior performance in comparison to state-of-the-art memory-based CL methods, and surpasses the state of the art with a large margin in the challenging settings of imbalanced classes, imbalanced domains, and imbalanced classes and domains scenarios. Furthermore, we experimentally show that the proposed **FDBS** for memory selection and **IWL** can be easily combined with state-of-the-art CL methods and improve their performance with significant margins.

## 2 RELATED WORK

**Continual learning**  Last years have seen significant progress in CL and recorded three major approaches: *Regularisation-based* methods (*e.g.*, Kirkpatrick et al. (2017); Zenke et al. (2017); Chaudhry et al. (2018)) impose regularization constraints on parameter changes to prevent forgetting previously learned knowledge. *Architecture-based* methods (*e.g.*, Serrà et al. (2018); Yan et al. (2021); Douillard et al. (2022); Ye & Bors (2022); Yan et al. (2021); Gao et al. (2023)) involve network isolation or expansion as strategies for enhancing network performance during continual learning *Memory-based* methods (*e.g.*, Rolnick et al. (2019); Aljundi et al. (2019b); Bang et al. (2021); Aljundi et al. (2019a); Yoon et al. (2022)) store or generate a small subset of the data samples from past tasks and then replay them during the current task training to retain past task knowledge. Nonetheless, the majority of these methods are typically evaluated using balanced datasets and are designed for the Class-Incremental Learning (CIL) paradigm. In CIL, mutually exclusive class boundaries are assumed, meaning data for a new class is introduced and learned only once within a single task. In contrast, the proposed MSCL is an online CL method dealing with non-stationary data streams.

**Task-Free online continual learning**  Aljundi et al. (2018); Rolnick et al. (2019) introduce a novel CL scenario where task boundaries are not predefined, and the model encounters data in an online setting. Several memory-based strategies have been proposed to navigate this scenario. Reservoir Sampling (**ER**) Rolnick et al. (2019) assigns an equal chance for each piece of data to be selected in an online setting. However, this method can be easily biased by imbalanced data stream in terms of class and/or domain and inadvertently miss data that are more representative. Maximally Interfered Retrieval (**MIR**)Aljundi et al. (2019a) makes use of **ER** for data selection but retrieves the samples from the memory set which are most interfered for current learning. Gradient-based Sample Selection (**GSS**) Aljundi et al. (2019b) proposes to maximize the variance of gradient directions of the data samples in the replay buffer for data sample diversity but with no guarantee that the selected data are class representative. Furthermore, the replay buffer can be quickly saturated without any further update when local maximum of gradient variance is achieved. Online Corset Selection (**OCS**) Yoon et al. (2022) also employs the model's gradients for cosine similarity computation to select informative and diverse data samples in affinity with past tasks. Unfortunately, they are not class aware and its effectiveness diminishes when handling imbalanced data. In contrast, our proposed MSCL makes use of FDBS to promote the selection of informative data samples in terms of intra-class diversity and inter-class similarity in the feature space for storage. It further improves discriminative data representation using a built-in contrastive loss IWL.

**Imbalanced continual learning**  Wu et al. (2019) highlighted the limitations of existing CL methods, such as iCaRL Rebuffi et al. (2017) and EEIL Castro et al. (2018), in handling a large number of classes. The authors attributed these shortcomings to the presence of imbalanced data and an increase in inter-class similarity. To address this, they proposed evaluating CL methods in an imbalanced class-incremental learning scenario, where the data distribution across classes varies ((also

known as Long-Tailed Class Incremental Learning, as defined by Liu et al. (2022))). In order to mitigate this issue, they introduced a simple bias correction layer to adjust the final output during testing. One approach described by Chrysakis & Moens (2020) is CBRS (Class-Balancing Reservoir Sampling), which is based on the reservoir sampling technique Vitter (1985). This algorithm assumes equal data storage for each category and employs reservoir sampling within each category. However, when faced with imbalanced domain-incremental learning scenarios where the data distribution within domains is uneven, CBRS can only perform random selection, limiting its effectiveness. Instead, our proposed MSCL performs dynamically class informed data sample selection.

**Contrastive learning in Continual learning** Continual learning methods(*e.g.*, Lange & Tuytelaars (2021); Mai et al. (2021); Wei et al. (2023)) utilizing contrastive learning primarily rely on supervised contrastive learning proposed by Khosla et al. (2021). These methods typically necessitate extensive data augmentation to enhance representation learning, yet they often neglect the memory selection process. In our approach, we avoid using data augmentation and instead integrate contrastive learning with our FDBS to obtain a more representative memory set and to improve the feature extractor.

## 3 PRELIMINARY AND PROBLEM STATEMENT

We consider the setting of online task-free continual learning. The learner receives non-stationary data stream $\mathbb{O}$ through a series of data batches denoted as $\mathbb{S}_t^{str} = (x_i, y_i)_{i=1}^{N_b}$ at time step $t$. Here, $(x_i, y_i)$ represents an input data and its label, respectively, and $N_b$ denotes the batch size. The learner is represented as $f(\cdot; \boldsymbol{\theta}) = g \circ F$, where $g$ represents a classifier and $F$ denotes a feature extractor. We define a memory set as $\mathbb{S}^{mem} = (x_j, y_j)_{j=1}^{M}$, where $M$ is the memory size. We use the function $l(\cdot, \cdot)$ to denote the loss function. The global objective from time step $0$ to $T$ can be computed as follows:

$$l^* = \sum_{t=0}^{T} \sum_{(x_i, y_i) \in \mathbb{S}_t^{str}} l(f(x_i; \boldsymbol{\theta}), y_i) \tag{1}$$

However, within the setting of online continual learning, the learner does not have access to the entire data at each training step but only the current data batch and those in the memory set if any memory. Therefore, the objective at time step $T$ can be formulated as follows:

$$l_T = \underbrace{\sum_{(x_i, y_i) \in \mathbb{S}_T^{str}} l(f(x_i; \boldsymbol{\theta}_{T-1}), y_i)}_{\text{current loss}} + \underbrace{\sum_{(x_j, y_j) \in \mathbb{S}^{mem}} l(f(x_j; \boldsymbol{\theta}_{T-1}), y_j)}_{\text{replay loss}} \tag{2}$$

As a result, to enable online continual learning without catastrophic forgetting, one needs to minimize the gap between $l^*$ and $l^T$:

$$\min(l^* - l_T) = \min(\sum_{t=0}^{T-1} \sum_{(x_i, y_i) \in \mathbb{S}_t^{str} \setminus \mathbb{S}^{mem}} l(f(x_i; \boldsymbol{\theta}_{T-1}), y_i)) \tag{3}$$

In this paper, we are interested in memory-based online CL. Our objective is to define a strategy which carefully selects data samples to store in the memory set and continuously refines data representation so as to minimize the gap as shown in Eq. (3).

## 4 METHODOLOGY

### 4.1 FEATURE-DISTANCE BASED SAMPLE SELECTION

In the context of imbalanced online domain and class continual learning scenarios, models need to contend with at least two types of distribution shifts: correlation shift and diversity shift. In classification problems, these distribution shifts can result in increased inter-class similarity and intra-class variance, ultimately leading to catastrophic forgetting. Current memory selection methods (e.g., ER

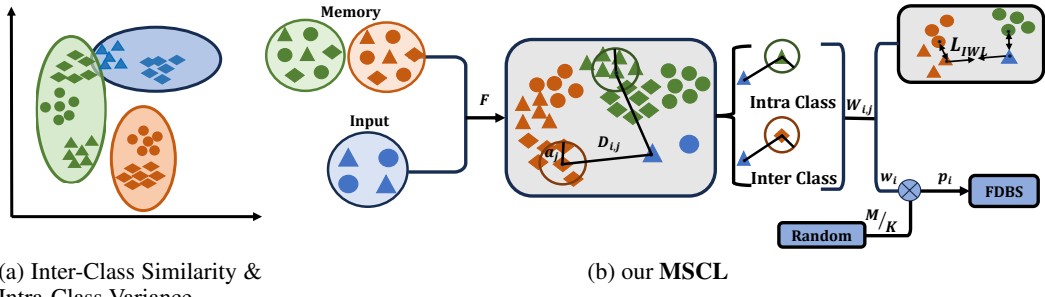

(a) Inter-Class Similarity &
Intra-Class Variance

(b) our **MSCL**

Figure 1: Both figures use colors to represent domains, while shapes distinguish between categories.**(a)**In practical continual learning scenarios, models must adapt to large-scale datasets characterized by both inter-class similarity and intra-class variance. In this illustration, the orange diamond is distantly related to the green diamond, while the blue triangle exhibits proximity to the green diamond. These disparities challenge the model's performance in continual learning.**(b)**Our proposed MSCL involves mapping input data and a memory set into a shared feature space. Here, $\boldsymbol{D}_{i,j}$ represents the distance between input data $x_i$ and data $x_j$ in the memory set. We use the same indexing convention for other formulas. We calculate distances, $\boldsymbol{D}$ and $\boldsymbol{a}$, between input data and memory set, and then derive an importance weight matrix quantifying each input data representative importance w.r.t those in the memory set based on the analysis of their intra-class diversity or inter-class similarity in the feature space. These importance weights are combined with random selection to give birth to our Feature-Distance based Sample Selection (FDBS) which identifies the most representative input data points for storage into the memory set. Armed with this importance weight matrix, we proceed to craft a novel Contrastive Loss (IWL) aimed at refining the feature space by compacting intra-class data and creating greater separation among inter-class data.

Rolnick et al. (2019), CBRS Chrysakis & Moens (2020), GSS Aljundi et al. (2019b), OCS Yoon et al. (2022)) are unable to effectively address both of these challenges simultaneously. To tackle this issue, we introduce our feature-based method, referred to as Feature-Based Dissimilarity Selection (FDBS). FDBS encourages the model to select data points that are the most dissimilar within a class and the most similar between different classes. This strategy aims to enhance both inter-class similarity and intra-class variance within the memory set. Consequently, FDBS helps to narrow the gap between the memory set and the true data distribution, as demonstrated in Equation 3.

Our proposed method, denoted as **FDBS**, is shown in Appendix Algorithm 2, with $M$ denoting the memory size and $K$ the number of data samples so far streamed. When the learner receives a batch of data $\mathbb{S}^{str}$ from the stream $\mathbb{O}$, we check for each new data sample $x_i$ in $\mathbb{S}^{str}$ whether the memory set is full. If it is not full, we can directly store $x_i$. However, if the memory set is full, we need to evaluate the importance weight $w_i$ of the new data sample $x_i$ to determine whether it is worth storing. The key to this process is to keep the memory set aware of intra-class diversity and inter-class boundaries based on the feature distances between the new data sample $x_i$ and the memory set. It involves the following three main steps:

- We begin by calculating the feature distance, denoted as $\boldsymbol{D}$ (refer to Eq. (4)), between every data point in the set $\mathbb{S}^{str}$ and each data sample stored in the memory set $\mathbb{S}^{mem}$. Subsequently, we identify the minimum distance between the input data and the memory set for each input data sample, resulting in the vector $\boldsymbol{d}^{str}$ as defined in Eq. (4)

$$\boldsymbol{D}_{i,j} = dist\left\{F(x_i), F(x_j)\right\}_{(x_i \in \mathbb{S}^{str}; x_j \in \mathbb{S}^{mem})} \quad ; \quad \boldsymbol{d}_i^{str} = min(\boldsymbol{D}_{i,:}) \tag{4}$$

- Subsequently, we compute $\boldsymbol{D}^{mem}$, as in Eq. (5), the feature distance between every pair of points in the memory set, and the minimum distance for each data point in the memory set in $\boldsymbol{d}^{mem}$, as shown in Eq. (5). We then calculate $\boldsymbol{a}$ as in Eq. (6) a weighted average distance from a data point in the memory set to all other points, using a RBF kernel as in Eq. (6) to weight the distances. We aim to assign higher weight to closer distances.

$$\boldsymbol{D}_{i,j}^{mem} = dist\left\{F(x_i), F(x_j)\right\}_{(x_i, x_j \in \mathbb{S}^{mem})} \quad ; \quad \boldsymbol{d}_i^{mem} = min(\boldsymbol{D}_{i,j\neq i}^{mem}) \tag{5}$$

- By computing the difference between $\boldsymbol{a}$ and $\boldsymbol{D}$, we can derive an **importance weight** for each new data. This weight is subsequently combined with the reservoir sampling coefficient to determine the probability of selecting the new data point.

$$\boldsymbol{\alpha}_{i,j} = e^{-\frac{\left\| \boldsymbol{D}_{i,j}^{mem} - \boldsymbol{d}_i^{mem} \right\|^2}{2\sigma^2}} \quad ; \quad \boldsymbol{a}_i = \frac{\sum_{j \neq i}^M \boldsymbol{D}_{i,j}^{mem} \boldsymbol{\alpha}_{i,j}}{\sum_{j \neq i}^M \boldsymbol{\alpha}_{i,j}} \tag{6}$$

**Importance weight** is the core concept of our proposed method. It serves to assess the significance of a new data sample with respect to the memory set, with a focus on promoting diversity among previously encountered intra-class data while also considering the potential closeness to inter-class boundaries. Specifically, we calculate this importance weight, as defined in Eq. (8), to capture the influence of each data point in the memory set on an input data sample. This influence is determined by whether they belong to the same class, as illustrated in Fig. 1 (b). Our approach is based on the intuitive notion that when two points, $x_i$ and $x_j$, are closer in proximity, the impact of $x_j$ on $x_i$ becomes more pronounced. To achieve this, we employ a Radial Basis Function (RBF) kernel, as expressed in Eq. (7). This kernel ensures that the influence of distant points diminishes rapidly. Additionally, we use the sign function, as shown in Eq. (7), to assign a value of 1 if the classes are the same and -1 otherwise.

When comparing a new data sample $x_i$ with a memory set data point $x_j$, we consider two scenarios based on their class labels. If they share the **same class label**, as shown in Fig. 1 (b), and if the feature distance $\boldsymbol{D}_{i,j}$ significantly exceeds $\boldsymbol{a}_j$, it implies a substantial difference between $x_i$ and $x_j$. In this case, we assign $\boldsymbol{W}_{i,j}$ a value greater than 1, promoting the selection of $x_i$ for storage. However, when $x_i$ and $x_j$ have **different class labels**, we aim to store data points near decision boundaries to capture closer class boundaries caused by increased inter-class similarities. We achieve this by setting $\boldsymbol{W}_{i,j}$ using Eq. (8) with the sign function returning -1. If $\boldsymbol{a}_j$ significantly surpasses $\boldsymbol{D}_{i,j}$, it implies that despite their different labels, $x_i$ closely resembles $x_j$, motivating us to store $x_i$. Conversely, if $\boldsymbol{a}_j$ is substantially smaller than $\boldsymbol{D}_{i,j}$, it suggests that the model can readily distinguish between $x_i$ and $x_j$, leading us to exclude $x_i$ from storage. When $\boldsymbol{D}_{i,j}$ is approximately equal to $\boldsymbol{a}_j$, we consider $x_i$ as a typical data point close to $x_j$, leading $\boldsymbol{W}_{i,j}$ to approach 1, resulting in a random selection.

$$\boldsymbol{\beta}_{i,j} = e^{-\frac{\left\| \boldsymbol{D}_{i,j} - \boldsymbol{d}_i^{str} \right\|^2}{2\sigma^2}} \quad ; \quad sgn(y_i, y_j) = \begin{cases} 1 \text{ if } y_i = y_j \\ -1 \text{ if } y_i \neq y_j \end{cases} \tag{7}$$

$$\boldsymbol{W}_{i,j} = e^{sgn(y_i, y_j) \frac{\boldsymbol{D}_{i,j} - \boldsymbol{a}_j}{\boldsymbol{D}_{i,j} + \boldsymbol{a}_j} \boldsymbol{\beta}_{i,j} \tau} (y_i \in \mathbb{S}^{str}; y_j \in \mathbb{S}^{mem}) \tag{8}$$

To take into account the influence of all data points in the memory set on a new input data point for its importance weight, we directly multiply the impact of each memory point as shown in Eq. (9).

To get the final probability $p_i$ for a new data sample $x_i$ to be chosen for storage in memory, we introduce the reservoir samplingRolnick et al. (2019). Given a fixed memory size $M$ and the number of data samples observed so far in the data stream, denoted as $K$, $M/K$ represents the probability of each data sample being randomly selected. We then use the importance weight $\boldsymbol{w}_i$ to adjust the probability of the new data sampled $x_i$ being selected as shown in Eq. (9). This allows us to handle imbalanced data and retain a certain level of randomness.

$$\boldsymbol{w}_i = \prod_{j=1}^M \boldsymbol{W}_{i,j} \quad ; \quad p_i = min(\boldsymbol{w}_i \frac{M}{K}, 1) \tag{9}$$

## 4.2 CONTRASTIVE LEARNING FOR BETTER DISCRIMINATIVE FEATURE REPRESENTATION

The importance weight $\boldsymbol{W}_{i,j}$, calculated using Eq. (8), quantifies the similarity between two data points in the feature space and is differentiable. Drawing inspiration from contrastive learning methods that aim to maximize similarity between positive pairs of samples and minimize similarity between negative pairsDong & Shen (2018); Schroff et al. (2015), we introduce a specialized contrastive learning loss (IWL) to refine our feature representation with the current data. Our IWL

is designed to reduce inter-class similarity and intra-class variance within the memory set, effectively acting as an adversarial component to our memory selection process. Additionally, it serves to compact the feature space of our memory set, facilitating more representative memory selection in subsequent operations. Specifically, for a batch of data with size $N_b$, we sample a minbatch data from the memory set with size $N_m$. The IWL is computed as in Eq. (10). Minimizing $\boldsymbol{W}_{i,j}$ will bring data points closer when their class labels are the same, while pushing them further apart when their class labels are different.

$$L_{IWL} = \frac{\sum_{i=1}^{N_m} \sum_{j=1}^{N_b} log(\boldsymbol{W}_{i,j})}{\sum_{i=1}^{N_m} \sum_{j=1}^{N_b} \boldsymbol{\beta}_{i,j}} \tag{10}$$

The final algorithm is presented in Appendix Algorithm 1. In our algorithm, to reduce computational complexity, we do not fully update $\boldsymbol{D}^{mem}$ at each step. Instead, during each iteration, we draw a small batch of data from the memory set and dynamically update the corresponding distances and feature vectors for that specific batch.

## 5 EXPERIMENTS

### 5.1 BALANCED BENCHMARKS

Building upon previous research van de Ven & Tolias (2019); Aljundi et al. (2019b); Douillard et al. (2020); Volpi et al. (2021), we utilize four well-established Continual Learning (CL) benchmarks: Split MNIST, Split ImageNet-1k, Split CIFAR-100, and PACS. Split MNISTDeng (2012) comprises five tasks, each containing two classes. For Split CIFAR-100, we partition the original CIFAR-100 dataset Krizhevsky (2009) into ten subsets, with each subset representing a distinct task comprising ten classes. For Split mini-ImageNetVinyals et al. (2016), we partition the original mini-ImageNet dataset Krizhevsky (2009) into ten subsets, with each subset representing a distinct task comprising ten classes. As for PACS Li et al. (2017), it encompasses four domains: photo, art painting, cartoon, and sketch. Each domain consists of the same seven classes. In our experiments, we treat each domain as an individual task, resulting in a total of four tasks. Notably, due to significant differences between images in each domain, one can observe a notable increase in inter-class variance within this dataset.

### 5.2 IMBALANCED BENCHMARKS

Previous CL benchmarks have roughly the same number of instances per class and domain and therefore cannot be used to benchmark CL methods on non-stationary data with imbalanced classes and/or domains. As a result, we have designed some specific benchmarks to highlight the robustness of CL methods with respect to imbalanced data.

**Imbalanced Class-Incremental Learning (Imb CIL).** To establish an imbalanced Class-incremental scenario for split CIFAR-100 and split mini-ImageNet, we build upon the approach introduced by Chrysakis & Moens (2020). Unlike traditional benchmarks that distribute instances equally among classes, we induce class imbalance by utilizing a predefined ratio vector, denoted as $\mathbf{r}$, encompassing five distinct ratios: $(10^{-2}, 10^{-1.5}, 10^{-1}, 10^{-0.5}, 10^0)$. In this setup, for each run and each class, we randomly select a ratio from $\mathbf{r}$ and multiply it by the number of images corresponding to that class. This calculation determines the final number of images allocated to the class, thus establishing our imbalanced class scenario. We maintain the remaining conditions consistent with the corresponding balanced scenario.

**Imbalanced Domain-incremental Learning (Imb DIL).** We adapt the PACS dataset, encompassing four domains, and follow an approach akin to our Imbalanced Class-Incremental method. For each domain, we randomly select a ratio from $\mathbf{r}$, multiply it with the image count of the domain, thereby maintaining a balanced class count within the imbalanced domain.

**Imbalanced Class and Domain Incremental Learning (Imb C-DIL).** We further refine the PACS dataset to generate an imbalanced class-domain incremental scenario, which mirrors a more

Table 1: We report the results of our experiments conducted on **balanced** scenarios. We present the final accuracy as mean and standard deviation over five independent runs. For Split CIFAR-100 and mini-ImageNet, the memory size was set to 5000, while for all other scenarios, the memory size was set to 1000.

| Methods / Datasets | Split MNIST | mini ImageNet | Split CIFAR-100 | PACS |
|---|---|---|---|---|
| Fine tuning | $19.23 \pm 0.32$ | $4.21 \pm 0.22$ | $4.43 \pm 0.17$ | $20.56 \pm 0.24$ |
| i.i.d. Offline | $92.73 \pm 0.21$ | $52.52 \pm 0.05$ | $49.79 \pm 0.28$ | $56.94 \pm 0.12$ |
| ER | $81.68 \pm 0.97$ | $15.76 \pm 2.34$ | $18.26 \pm 1.78$ | $41.66 \pm 1.45$ |
| GSS | $80.38 \pm 1.42$ | $12.31 \pm 1.26$ | $13.57 \pm 1.23$ | $39.87 \pm 3.25$ |
| CBRS | $81.34 \pm 1.27$ | $15.58 \pm 1.94$ | $18.55 \pm 1.68$ | $41.34 \pm 1.65$ |
| MIR | $\mathbf{86.76} \pm 0.67$ | $16.73 \pm 1.12$ | $18.71 \pm 0.89$ | $42.2 \pm 0.85$ |
| OCS | $85.43 \pm 0.86$ | $16.59 \pm 0.89$ | $19.31 \pm 0.48$ | $42.63 \pm 0.73$ |
| **FDBS** | $85.79 \pm 0.76$ | $17.54 \pm 2.17$ | $19.89 \pm 1.54$ | $42.86 \pm 1.37$ |
| **FDBS+IWL** | $86.48 \pm 0.57$ | $\mathbf{18.93} \pm 0.74$ | $\mathbf{21.13} \pm 0.94$ | $\mathbf{43.54} \pm 0.75$ |

Table 2: Results on our **imbalanced** scenarios. We present the final accuracy as mean and standard deviation over five independent runs. For PACS, the memory size was set to 1000, while for all other scenarios, the memory size was set to 5000.

| Scenarios | **Imb CIL** | | **Imb DIL** | **Imb C-DIL** | |
|---|---|---|---|---|---|
| | CIFAR-100 | mini-ImageNet | PACS | PACS | DomainNet |
| Fine Tunning | $3.18 \pm 0.31$ | $3.57 \pm 0.25$ | $15.54 \pm 1.34$ | $14.35 \pm 1.23$ | $2.35 \pm 0.65$ |
| i.i.d. Offline | $41.65 \pm 0.57$ | $43.17 \pm 0.62$ | $46.34 \pm 0.47$ | $46.18 \pm 0.92$ | $37.27 \pm 0.73$ |
| ER | $7.14 \pm 0.81$ | $8.25 \pm 1.27$ | $25.64 \pm 2.19$ | $22.48 \pm 1.23$ | $6.24 \pm 0.62$ |
| GSS | $8.38 \pm 0.74$ | $7.95 \pm 0.48$ | $24.46 \pm 1.78$ | $20.17 \pm 2.14$ | $5.15 \pm 0.44$ |
| CBRS | $10.21 \pm 0.39$ | $11.37 \pm 0.63$ | $25.97 \pm 1.54$ | $23.68 \pm 1.75$ | $6.13 \pm 0.59$ |
| MIR | $7.52 \pm 0.93$ | $8.97 \pm 0.30$ | $25.85 \pm 2.19$ | $22.15 \pm 2.57$ | $6.47 \pm 0.45$ |
| OCS | $11.68 \pm 0.63$ | $12.29 \pm 0.49$ | $27.15 \pm 1.42$ | $24.72 \pm 1.37$ | $8.47 \pm 0.78$ |
| **FDBS** | $12.35 \pm 0.85$ | $12.89 \pm 0.62$ | $29.13 \pm 1.53$ | $27.56 \pm 1.52$ | $10.25 \pm 0.94$ |
| **FDBS+IWL** | $\mathbf{13.72} \pm 0.53$ | $\mathbf{14.21} \pm 0.34$ | $\mathbf{31.25} \pm 0.83$ | $\mathbf{28.64} \pm 1.44$ | $\mathbf{11.46} \pm 0.71$ |

realistic data setting. This scenario involves randomly selecting a ratio from **r** for each class and domain, and multiplying it with the count of instances for that class within the domain. This operation yields $4 * 7$ values for PACS, resulting in a diverse number of data points across different classes and domains. This approach accentuates the growth of inter-class similarity and intra-class variance. Because both the class and domain are already imbalanced in the original **DomainNet**Peng et al. (2019), we directly use its original format to generate the imbalanced scenario. We adhere to a sampling without replacement strategy for data stream generation. Once data from a pair of class and domain is exhausted, we transition to the next pair.

## 5.3 BASELINES AND IMPLEMENTATION DETAILS

As the proposed FDBS is a memory-based online CL method, we compare it primarily against other memory-centric techniques such as Experience Replay (ER) Rolnick et al. (2019), Gradient-Based Sample Selection (GSS) Aljundi et al. (2019b), Class-Balancing Reservoir Sampling (CBRS) Chrysakis & Moens (2020), Maximally Interfering Retrieval (MIR) Aljundi et al. (2019a), and Online Corset Selection(OCS)Yoon et al. (2022).

We include Fine-tuning (FT), the process of utilizing preceding model parameters as initial parameters for the subsequent task without a memory set, as a lower bound for comparison. In contrast, i.i.d. offline training represents a formidable upper bound as it provides the learner with access to the complete dataset for model training, rather than a sequential stream of batches. This approach holds a significant advantage by allowing the learner to iterate over the entire training data for multiple epochs, maximizing its potential performance. Our proposed strategy comprises two key components: Feature-Distance Based Sampling Selection (FDBS) for sample selection and Contrastive

Learning Loss (IWL) for discriminative representation learning. We evaluate the efficacy of using FDBS solely and in conjunction with IWL in our experiments.

**Implementation details.** For MNIST, we utilize a two-hidden-layer MLP with 250 neurons per layer. Meanwhile, for all other datasets, we adopt the standard ResNet-18 He et al. (2016)architecture implemented in PyTorchPaszke et al. (2019). The replay buffer size is configured as 5000 for CIFAR-100, mini-ImageNet, and DomainNet, while it is set to 1000 for all other scenarios. We maintain a fixed batch size of 20 for the incoming data stream, with five update steps per batch. Notably, we abstain from employing data augmentation in our experiments. We utilize the Adam optimizer Kingma & Ba (2015), set the $\sigma$ value in our radial basis function (RBF) kernel at 0.5, and the $\tau$ value in Eq. (8) at 0.5. Our approach's performance is evaluated across the balanced and imbalanced benchmarks through five independent runs, from which we compute the average accuracy.

## 6 RESULTS

The effects of memory size on our FDBS method are detailed in Appendix A.2 and presented in Table Tab. 3. Furthermore, the utilization of our proposed contrastive learning loss to enhance other state-of-the-art methods is discussed in Appendix A.5 and in Appendix A.3. The results on the classic class-incremental learning is detailed in Appendix A.7. An ablation study of hyperparameters is conducted in Appendix A.4, while an examination of the memory set distribution is presented in Appendix A.6.

### 6.1 RESULTS ON BALANCED BENCHMARKS

Results for balanced scenarios are shown in Tab. 1. While the Experience Replay (ER) method fares well in these settings due to its unbiased memory selection, our proposed FDBS method paired with the Contrastive Learning Loss (IWL) offers notable improvements. This enhancement is largely attributed to IWL's feature space optimization, which aids FDBS's data sample selection based on feature space distance. The combination of FDBS and IWL also yields more consistent results, as evidenced by a reduced standard deviation. Especially for datasets like Rotated MNIST and PACS, FDBS excels by augmenting intra-class diversity in memory, thus increasing adaptability to domain shifts.

### 6.2 RESULTS ON IMBALANCED SCENARIOS

Tab. 2 displays results in imbalanced settings. For imbalanced CIL scenarios, the CBRS method, which maintains an equal count of images from each class in memory, outperforms the basic ER approach. Meanwhile, OCS, by continuously evaluating data batch gradients, filters noise and selects more representative data, shining particularly in imbalanced contexts. However, our FDBS method stands out, consistently leading in all imbalanced tests. As scenarios evolve from Imb DIL to Imb C-DIL, other methods' accuracy drops significantly, but FDBS maintains robust performance. Its strength lies in using feature-distance to fine-tune memory selection, preserving class boundaries and boosting intra-class diversity. This advantage is amplified when paired with the IWL, reinforcing the benefits seen in balanced scenarios.

## 7 CONCLUSION

This paper presents a new online Continual Learning (CL) method, MSCL, consisted of Feature-Distance Based Sample Selection (FDBS) and Contrastive Learning Loss (IWL). FDBS selects representative examples by evaluating the distance between new and memory-set data, emphasizing dissimilar intra-class and similar inter-class data, thus increasing memory awareness of class diversity and boundaries. IWL minimizes intra-class and maximizes inter-class distances, enhancing discriminative feature representation. Extensive experiments confirmed that FDBS and IWL together outperform other memory-based CL methods in balanced and imbalanced scenarios. Future work will explore combining MSCL with a distillation-based CL method to further improve its performance.

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

## A  APPENDIX

You may include other additional sections here.

### A.1  CLARIFICATIONS OF THREE SCENARIOS

- **Task-Incremental Learning (TIL)**: This is a continual learning scenario where the model is informed about the task that needs to be performed in advance. In this scenario, the model can be trained with task-specific components as it knows what it's being asked to do. A typical architecture for a model in this scenario is a multi-headed output layer, meaning each task has its own output units, while the rest of the network may be shared between tasks. The goal is to incrementally improve on a series of tasks, learning each one without forgetting the previous tasks.

- **Domain-Incremental Learning (DIL)**: In this scenario, the model does not know the task identity at test time. However, it only needs to solve the task at hand, without necessarily identifying which task it is. The structure of tasks remains consistent, but the input distribution may vary. The model needs to adapt to these changes in the input distribution to

successfully perform the task. A real-world example might be a model learning to adapt to different environments without explicitly identifying the environment.

- **Class-Incremental Learning (CIL)**: This is a complex learning scenario where the model not only needs to solve each task it has encountered so far but also must infer which task it is currently facing. In other words, it should be able to classify and learn new classes of objects incrementally. The model is required to maintain knowledge of previously learned classes while still being able to learn new ones. This scenario embodies many real-world learning problems where new classes or categories are continually encountered, and old ones should not be forgotten.

## A.2    RESULTS ON DIFFERENT MEMORY SIZES

To evaluate the performance of our proposed method under varying memory sizes, we conducted experiments by adjusting the size of the memory set and comparing the results with those obtained using other memory selection methods. The experiments were conducted using the imbalanced class-domain incremental scenario of PACS, and the results are presented in Tab. 3.

The experimental results showed consistent performance improvements for our proposed FDBS method across all memory sizes tested. Our method outperformed all other memory selection methods in each case, with the magnitude of the improvement being more pronounced for smaller memory sizes. Furthermore, our proposed FDBS method can be further strengthened by combining it with Contrastive Learning Loss (IWL) to improve its performance

Table 3: Comparison of different memory selection methods on Imb C-DIL PACS for three different memory sizes. We present the final accuracy as mean and standard deviation over five independent runs

| Methods | Memory size | | |
|---|---|---|---|
| | 100 | 500 | 2000 |
| ER | 16.47±2.39 | 20.34±2.56 | 24.37±1.34 |
| GSS | 15.73±1.63 | 17.67±1.95 | 23.28±1.39 |
| CBRS | 17.24±2.15 | 21.15±2.17 | 25.61±1.84 |
| OCS | 19.35±1.87 | 23.43±2.28 | 26.87±1.36 |
| **FDBS**(ours) | 19.76±1.96 | 25.56±2.61 | 29.12±2.48 |
| **FDBS+IWL**(ours) | **21.22**±1.48 | **26.34**±1.86 | **30.28**±0.93 |

## A.3    THE EFFECTIVENESS OF IWL

We combined Contrastive Learning Loss (IWL) with ER and CBRS to evaluate the effectiveness of our IWL. The experiments were conducted on Imbalanced C-DIL DomainNet and the Balanced CIFAR-100. The results are presented in Tab. 4.

Our study has demonstrated that Contrastive Learning Loss (IWL) can significantly enhance the performance of simple memory sample selection methods. Specifically, IWL is capable of optimizing the feature space, thereby enabling model better classifying. Additionally, we have observed that our selection method, FDBS, achieves the best results when used in combination with IWL.

## A.4    STUDY THE INFLUENCE OF HYPERPARAMETERS

In our memory selection method, FDBS, we introduce two crucial hyperparameters: $\sigma$ within the RBF kernel (Eq. (7)) and $\tau$ as defined in (Eq. (8)). To assess the impact of these hyperparameters, we conducted experiments specifically on the Imbalanced Class-Domain Incremental Learning (Imb C-DIL) scenario of PACS. The results of these experiments are presented in Appendix A.4.

In our approach, both $\sigma$ and $\tau$ play pivotal roles in evaluating the influence of a memory point on an input point, based on their respective distances. Generally, a larger value for these hyperparameters signifies that the influence diminishes more rapidly as the distance between points increases. Through our experimentation, we observed that our model exhibits a higher sensitivity to variations in the value of $\tau$ than $\sigma$.

Table 4: We combined Contrastive Learning Loss (IWL) with ER and CBRS to evaluate the effectiveness of our IWL. The experiments were conducted on Imbalanced C-DIL DomainNet and Balanced CIFAR-100. We set the memory size as 5000. The final accuracy was presented as the mean and standard deviation over five independent runs

| Methods/Datasets | Balanced CIFAR-100 | Imb C-DIL DomainNet |
|---|---|---|
| ER | $18.26 \pm 1.78$ | $6.24 \pm 0.62$ |
| ER+IWL | $18.79 \pm 1.32$ | $8.34 \pm 0.54$ |
| CBRS | $18.55 \pm 1.68$ | $6.13 \pm 0.59$ |
| CBRS+IWL | $19.13 \pm 1.16$ | $9.21 \pm 0.63$ |
| FDBS | $19.89 \pm 1.54$ | $10.25 \pm 0.94$ |
| FDBS+IWL | $\mathbf{21.13} \pm 0.94$ | $\mathbf{11.46} \pm 0.71$ |

| $\tau$ | $\sigma = 0.5$ |
|---|---|
| 0.1 | $27.23 \pm 1.89$ |
| 0.5 | $\mathbf{28.64} \pm 1.44$ |
| 1 | $27.58 \pm 1.46$ |
| 5 | $26.18 \pm 1.23$ |
| 10 | $24.89 \pm 1.13$ |

| $\sigma$ | $\tau = 0.5$ |
|---|---|
| 0.1 | $28.50 \pm 1.65$ |
| 0.5 | $\mathbf{28.64} \pm 1.44$ |
| 1 | $28.34 \pm 1.32$ |
| 5 | $28.2 \pm 1.35$ |
| 10 | $27.49 \pm 1.26$ |

Table 5: $\sigma$ fixed while varying $\tau$      Table 6: $\tau$ fixed while varying $\sigma$

## A.5 COLLABORATIVE LEARNING WITH OTHER MEMORY-BASED METHODS

In our evaluation, we consider two notable continual learning methods, PodNetDouillard et al. (2020) and AFCKang et al. (2022), both of which incorporate specialized distillation techniques reliant on a memory set. We integrate our Feature-Distance Based Sample Selection (FDBS) method to replace their original selection methods, which were either random or based on herding. Our experiments encompass two distinct scenarios: Balanced CIFAR-100 and the imbalanced Class-Domain Incremental Learning (imb C-DIL) of DomainNet. The results of these experiments are presented in Table Tab. 7. Remarkably, our memory selection method consistently enhances the performance of these continual learning methods both on balanced and imbalanced scenarios.

| Methods | Split-CIFAR100 | Imb C-DIL DomainNet |
|---|---|---|
| PodNet | $19.57 \pm 1.48$ | $8.75 \pm 0.73$ |
| PodNet + FDBS(ours) | $\mathbf{20.93} \pm 1.72$ | $10.32 \pm 0.82$ |
| AFC | $19.43 \pm 1.67$ | $7.69 \pm 0.64$ |
| AFC + FDBS(ours) | $20.69 \pm 1.54$ | $\mathbf{10.65} \pm 0.49$ |

Table 7: Combining FDBS with Other Memory-Based Methods: Experiments on Balanced Split CIFAR-100 and Imbalanced Class-Domain Incremental Learning on DomainNet (Memory Size: 5000).The final accuracy was presented as the mean and standard deviation over five independent runs.

## A.6 THE DISTRIBUTION OF OUR MEMORY SET

To gain deeper insights into the efficacy of our memory selection method, we examine the distribution of our memory set. Our experiments focus on the challenging task of imbalanced Domain-Incremental Learning using the PACS dataset, which comprises four distinct domains (e.g., photo, art painting, cartoon, and sketch). Following training, we analyze the distribution of our memory set, shedding light on how our method has shaped the representation of critical data points within this dynamic learning environment. The results of this analysis are presented in Tab. 8. And the ratio of different domain is shown in Fig. 2.

Methods such as ER and CBRS opt for random image selection, aiming to maintain a distribution akin to the original dataset. In contrast, our method prioritizes increasing intra-class diversity, thereby influencing a more balanced distribution of stored images. This approach plays a crucial

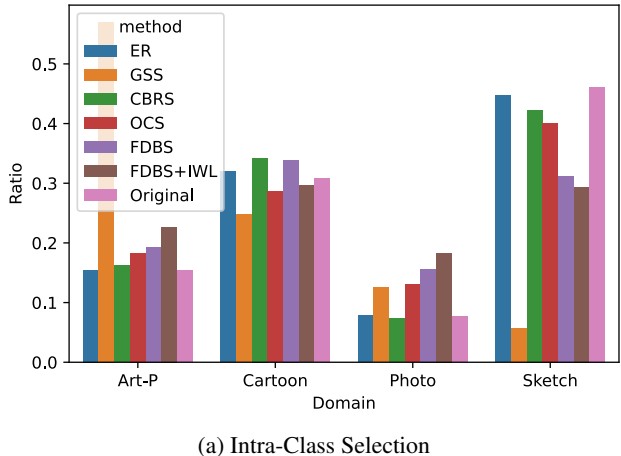

(a) Intra-Class Selection

Figure 2: The ratio of different domains within the memory set compared to the original scenario.

role in improving the overall performance of continual learning. Additionally, the integration of our Contrastive Learning Loss (IWL) further enhances the feature space within our memory set. This refinement proves instrumental in effectively capturing images from minority domains, contributing to a more robust and balanced representation of data.

| Methods /Domains | Photo | Art Painting | Cartoon | Sketch |
|---|---|---|---|---|
| Our Scenario | 500 | 1000 | 2000 | 3000 |
| ER | 78 | 155 | 320 | 447 |
| GSS | 125 | 570 | 248 | 57 |
| CBRS | 73 | 162 | 342 | 423 |
| OCS | 130 | 183 | 286 | 401 |
| FDBS(ours) | 156 | 193 | 339 | 312 |
| FDBS+IWL(Ours) | 183 | 227 | 296 | 294 |

Table 8: Comparison of Memory Set Composition Across Methods in Imbalanced Domain-Incremental Learning (imb DIL) Scenario of PACS. We set the memory size as 1000.

### A.7 RESULTS ON BALANCED CLASS-INCREMENTAL LEARNING SCENARIO

To assess the effectiveness of our proposed approach in the context of classic balanced class-incremental learning, we conducted an experiment referred to **Cifar 100-B0** as detailed in Yan et al. (2021). In this experiment, we partitioned the original Cifar 100 dataset into 10 and 20 distinct tasks, with each task encompassing a set of 5 distinct classes. The memory size is set as 2000. The result is presented in Tab. 9. Even in the classic class-incremental learning scenario, our proposed method can still significantly improve the previous state-of-the-art method.

| Methods | 10 steps | 20 steps |
|---|---|---|
| iCaRL*Rebuffi et al. (2017) | $65.27 \pm 1.02$ | $61.20 \pm 0.83$ |
| BiC*Wu et al. (2019) | $68.80 \pm 1.20$ | $66.48 \pm 0.32$ |
| PodNet*Douillard et al. (2020) | $58.03 \pm 1.27$ | $53.97 \pm 0.85$ |
| AFCKang et al. (2022) | $61.25 \pm 1.38$ | $54.76 \pm 0.79$ |
| WA*Zhao et al. (2019) | $69.46 \pm 0.29$ | $67.33 \pm 0.15$ |
| WA + **FDBS**(ours) | $\mathbf{71.35} \pm 0.56$ | $\mathbf{70.18} \pm 0.38$ |
| WA + **MSCL**(ours) | $\mathbf{73.71} \pm 0.27$ | $\mathbf{72.34} \pm 0.19$ |

Table 9: Results for classic class-incremental learning on CIFAR-100. Results marked with '*' are obtained directly from Yan et al. (2021). The memory size is set to 2000.

## A.8 ALGORITHM OF OUR METHOD

---

**Algorithm 1** Train a batch at time step t

---

**Input:** $F$, $g$ ,$\mathbb{S}^{mem}$, $\mathbb{S}_t^{str}$, $b$, $K$, $\boldsymbol{D}^{mem}$ as shown in Eq. (5), $\mathbf{F}^{mem}$ stores the features of the memory set, $N_b$ is the batch size.

1: **for** $b$ steps **do**
2:     sample batch $I, \mathbf{X}^m, \boldsymbol{y}^m$ of size $N_b$ from $\mathbb{S}^{mem}$ $\{I :$ the index of the samples in $S^{mem}\}$
3:     $\mathbf{X}^{str}, \boldsymbol{y}^{str} = \mathbb{S}_t^{str}$
4:     $\mathbf{F}^m, \hat{\boldsymbol{y}}^m = g \circ F(\mathbf{X}^m)$
5:     $\mathbf{F}^{str}, \hat{\boldsymbol{y}}^{str} = g \circ F(\mathbf{X}^{str})$
6:     $\alpha = 0.1 + 0.9 * 0.99^t$
7:     Current Loss : $L_{cur} = \ell(\hat{\boldsymbol{y}}^{str}, \boldsymbol{y}^{str})$
8:     Replay Loss : $L_r = \ell(\hat{\boldsymbol{y}}^m, \boldsymbol{y}^m)$
9:     Update $\mathbf{F}^{mem}[I] = \mathbf{F}^m$
10:     Update $\boldsymbol{D}^{mem}[I] = dist(\mathbf{F}^m, \mathbf{F}^{mem})$
11:     Compute $\boldsymbol{a}$ based on Eq. (6)
12:     $\boldsymbol{D} = dist(\mathbf{F}^{str}, \mathbf{F}^{mem})$ as Eq. (4)
13:     Compute $\boldsymbol{w}$ based on Eq. (8) and Eq. (9)
14:     $L_{IWL} = L_{IWL}(\boldsymbol{w})$ as Eq. (10)
15:     Total Loss : $L = \alpha L_{cur} + (1 - \alpha)L_r + L_{IWL}$
16:     Update: $F, g$ : Adam.step( )
17:     FDBS($\mathbb{S}^{mem}$,$\mathbb{S}_t^{str}$,$\boldsymbol{w}$,$\boldsymbol{D}$,$M$,$K$,$\boldsymbol{D}^{mem}$,$\mathbf{F}^{mem}$) as shown in Algorithm 2
18: **end for**

---

**Algorithm 2 FDBS** at time step t

---

**Input:** $\mathbb{S}^{mem}, \mathbb{S}_t^{str}, \boldsymbol{w}, \boldsymbol{D}, M, K, \boldsymbol{D}^{mem}, \mathbf{F}^{mem}$

1: $\mathbf{X}^{mem}, \boldsymbol{y}^{mem} = \mathbb{S}^{mem}$;
2: **for** each data $i, (x_i, y_i)$ in $\mathbb{S}_t^{str}$ **do**
3:     $K = K + 1$
4:     **if** $len(\mathbb{S}^{mem}) < M$ **then**
5:         store $(x_i, y_i)$ in $\mathbb{S}^{mem}$
6:     **else**
7:         $p = \boldsymbol{w}_i * M/K$
8:         $r = random.rand()$
9:         **if** $r < p$ or $y_i \notin \mathbb{S}^{mem}$ **then**
10:             $c = most\_frequent(\boldsymbol{y}^{mem})$
11:             $I = index(\boldsymbol{y}^{mem} == c)$
12:             $k = random.choice(I)$
13:             $\mathbf{X}^{mem}[k], \boldsymbol{y}^{mem}[k] = x_i, y_i$;
14:             $\mathbf{F}^{mem}[k] = F(x_i)$
15:             $\boldsymbol{D}^{mem}[k] = \boldsymbol{D}[i, :]$
16:         **else**
17:             ignore $(x_i, y_i)$
18:         **end if**
19:     **end if**
20: **end for**

---

