# OpenReview forum: "Imbalanced data robust online continual learning based on evolving class aware memory selection and built-in contrastive representation learning"
_ICLR.cc/2024/Conference — ICLR 2024 Conference Withdrawn Submission_

### Official Review · Reviewer_Zde2 · 2023-10-31

**Soundness:** 3 good
**Presentation:** 3 good
**Contribution:** 3 good
**Rating:** 5
**Confidence:** 3

**Summary:**

The paper introduces a novel Continual Learning (CL) algorithm, Memory Selection with Contrastive Learning (MSCL). It focuses on adapting to imbalanced data, shifting classes, and evolving domains. MSCL uses Feature-Distance Based Sample Selection (FDBS) to assess the representability of new data in memory, maintaining awareness of evolving inter-class similarities and intra-class diversity. Additionally, it incorporates a contrastive learning loss (IWL) that encourages similar class instances to be closer while pushing different class instances apart. Experimental results show that MSCL excels in challenging scenarios, enhancing existing CL methods in both balanced and imbalanced data settings.

**Strengths:**

1. This paper considers the setting of IMBALANCED DATA ONLINE CONTINUAL LEARNING that most online continual learning methods do not address.
2. Experimental results indicate that the proposed feature-distance-based sample selection and IWL are effective.
3. This paper conducts extensive experiments.

**Weaknesses:**

1. The proposed method appears to be more general rather than specifically tailored to the task of online continual learning with imbalanced data. Further clarification is needed regarding the relationship between the method and its motivation.
2. Computing the distance of all samples at each time step is time-consuming, which may not be practical for efficient online continual learning.
3. The method focuses on identifying samples with high intra-class variance and high inter-class similarity, potentially causing the model to overemphasize challenging extreme cases, whereas most samples exhibit low intra-class variance and high inter-class variance (class dissimilarity).
4. Typically, continual learning experiments provide accuracy changes for each incremental step. It would be beneficial to observe the accuracy changes for each task.
5. The baseline methods used for comparison seem somewhat outdated, with the most recent one dating back to 2022.

**Questions:**

typos. As a result, they face significant challenges in presence of imbalanced data in class and domain Wu et al. (2019)Liu et al. (2022). Ye et al. (2022) introduce a novel approach for quantifying dataset distribution shifts across two distinct dimensions.

---

> ### Author Response · Authors · 2023-11-21
>
> Thank you for your valuable suggestion.
>
> $Q1$: **The proposed method appears to be more general rather than specifically tailored to the task...** Our method is specifically engineered for the nuanced demands of online continual learning, especially in scenarios with imbalanced data. It addresses critical challenges such as shifting class dynamics and evolving domain characteristics, which are prevalent in such environments. our method incorporates a novel memory selection strategy, named Feature-Distance Based Sample Selection (FDBS), which effectively evaluates and adapts to the changing representational significance of new data in relation to the existing memory set. This approach is crucial for maintaining a relevant and adaptive memory set in the face of continuous data evolution. Additionally, the integration of a unique Contrastive Loss (IWL) aids in refining the feature space, enhancing the model's ability to compact intra-class data while distinctly separating inter-class data. This dual strategy of memory selection and feature space refinement is what sets MSCL apart, making it a tailored solution for the complexities of online continual learning in imbalanced data scenarios. In particular, our proposed memory selection strategy FDBS can be associated with any state of the art rehearsal-based CL methods to improve its performance and robustness to unbalanced data streams.
>
> $Q2$: **Computing the distance of all samples at each time step is time-consuming...** please see the response $Q1$ to Reviewer b5Cn.
>
> $Q3$: **The method focuses on identifying samples with high intra-class...** We acknowledge the validity of your comment regarding our method's approach to memory selection. Our strategy focuses on retaining samples that exhibit high intra-class variance and inter-class similarity, effectively capturing the most distinctive features of each class. However, this approach might inadvertently prioritize samples that are outliers or far from the data center, potentially leading to an over-representation of out-of-distribution (OOD) data in the memory set. This scenario could adversely impact the model's performance, especially in datasets with a significant presence of OOD data.
>
> To mitigate this issue, we implemented a comparison with random selection in our methodology, serving as a control to assess the effectiveness of our approach in managing OOD data. We recognize that distinguishing between OOD data and essential new information is a complex yet critical aspect of continual learning. This challenge underscores the need for continual learning models to be adept at discerning and adapting to the evolving nature of data distributions, effectively balancing the retention of valuable information while minimizing the influence of irrelevant data.
>
> $Q4$: **The baseline methods used for comparison seem somewhat outdated, with the most recent one dating back to 2022.** Our method primarily focuses on the management of the replay buffer which is the cornerstone of rehearsal-based CL methods and the underlying memory selection strategy is fundamental to deal with non stationary data streams. As such, the proposed memory selection strategy FDBS is orthogonal to state of the art rehearsal-based CL methods as it can be combined with any of them to improve their performance and robustness to unbalanced data. This is why we have chosen to compare it mainly with other memory selection methods. Furthermore, in A.7 (p15) of our submission as well as in our reply to Reviewer ddu2, we have further showcased the benefits of the proposed memory selection strategy when combined with two state of the art rehearsal-based CL methods in improving their performance and robustness to unbalanced data. Therefore, we believe the methods selected for comparison are representative and effectively demonstrate the efficacy of our approach.

---

### Official Review · Reviewer_b5Cn · 2023-11-01

**Soundness:** 3 good
**Presentation:** 3 good
**Contribution:** 2 fair
**Rating:** 5
**Confidence:** 3

**Summary:**

The paper introduces a online Continual Learning (CL) algorithm named Memory Selection with Contrastive Learning (MSCL), aiming to adaptively learn and retain knowledge in dynamic environments with imbalanced data. The proposed framework, MSCL, addresses the challenges arising from changes in correlations and diversities of data by continually adjusting previously established class and data representations. The core of MSCL lies in its two main components: Feature-Distance Based Sample Selection (FDBS) and Built-in Contrastive Learning Loss (IWL). The method has been tested on various datasets such as MNIST and Cifar-100. The results show that in balanced data scenarios, MSCL either matches or outperforms leading memory-based CL techniques, marginally. Additionally, the integration of FDBS and IWL enhances the performance of existing rehearsal-based CL methods in both balanced and imbalanced scenarios.

**Strengths:**

* Novel framework for online continual learning. The paper introduces Memory Selection with Contrastive Learning (MSCL), a novel framework specifically designed for online continual learning in dynamic environments with imbalanced data, shifting classes, and evolving domains.

* The paper tackles real-world challenges associated with non-stationary data streams, such as imbalanced data across classes and domains, and the need for ongoing adjustments in class and data representations.

**Weaknesses:**

* Lack of visualizations that show how data points are distributed in the feature space leaves the reader without a clear, visual understanding of how the Feature-Distance Based Sample Selection (FDBS) method operates.

* There is no discussion of the computational cost of the algorithm.

* Lack of discussions on the framework's applicability and performance in large-scale scenarios, which remain unaddressed in the current version of the paper.

**Questions:**

* There is no discussion of the computational cost of the algorithm. There are a few steps in the framework including: 1. Memory Management, 2. Feature Space Mapping, 3. Distance Calculation, 4. Importance Weight Calculation that can add to the computational cost hence overhead of the framework at each step. For example, the FDBS method involves managing a memory set, which includes calculating distances in the feature space and selecting representative samples. The size of the memory set and the dimensionality of the feature space can influence the computational cost and runtime efficiency for this step. The same applies to other steps.

* The authors mentioned Split ImageNet-1k in section 5. Experiment as one of the benchmarks but there are no mention of this dataset in the result section.

* The above comment raises another question: The authors acknowledge the substantial difficulties encountered when dealing with imbalanced data across various classes and domains, citing works by Wu et al. (2019) to underscore this challenge. However, the evaluations suffices to mid-sized datasets. This raises questions about the framework's applicability and performance in large-scale scenarios, which remain unaddressed in the current version of the paper.

* By continuously refining the feature space and adjusting the memory set based on incoming data, the method aims to adapt to changes in the data distribution. However, the paper could provide more details on how the method performs in scenarios with rapid changes in data distribution and whether there are any limitations in its adaptability.

* Lack of visualizations that show how data points are distributed in the feature space leaves the reader without a clear, visual understanding of how the Feature-Distance Based Sample Selection (FDBS) method operates.

---

> ### Author Response · Authors · 2023-11-21
>
> Thank you for your valuable suggestion.
>
> $Q1$ : **There is no discussion of the computational cost of the algorithm.** Our approach primarily incurs computational costs in calculating the matrix distance. For each input batch, it's necessary to compute the distance between the batch and the memory set. To optimize this process, our method employs the following strategies:
>
> - **Efficient Distance Matrix Computation:** We do not recalculate the entire distance matrix \( D \) (the distance between memory points) for each input data. Instead, we update only the parts associated with retrieved memory points. This approach significantly reduces the computational overhead.
>
> - **Linear Memory Cost:** The memory cost scales linearly with the size of the memory set, avoiding exponential increases.
>
> - **Computational Cost Evaluation:**
>
> To provide a clearer picture of the computational efficiency, we conducted tests on a single RTX3070 using ResNet18 with a memory size of 1000 in a balanced PACS setting. The results are as follows:
>
> | Method | Training Time (hours) |
> | ------ | --------------------- |
> | ER     | 0.18                  |
> | GSS    | 0.85                  |
> | OCS    | 0.26                  |
> | Ours   | 0.68                  |
>
> As illustrated in the table, our method requires more computation time than ER and OCS, but is less computationally intensive than GSS. This positions our method as a viable option in terms of computational efficiency, particularly when considering the enhanced performance it offers in more complex and realistic learning scenarios.
>
> $Q2$ : **The authors mentioned Split ImageNet-1k**. Sorry, there was a typo error. It should be the split mini-ImageNet.
>
> $Q3$ : **The authors acknowledge the substantial difficulties encountered when dealing with imbalanced data across various classes ...**
> Our method's adaptability to both balanced and imbalanced scenarios, and its scalability in terms of memory size, demonstrate its potential applicability in large-scale scenarios. Although we did not explicitly test on extremely large datasets due to resource constraints, the principles underpinning our method – particularly the efficient memory management and contrastive learning components, are designed to be scalable and applicable in large-scale settings. The scalability of our method is demonstrated by its application from the smaller PACS dataset to the larger DomainNet dataset.
>
> $Q4$: **By continuously refining the feature space and adjusting the memory set based on incoming data...**  In our research, we utilized the PACS dataset, which comprises four distinct domains: Photo, Art-painting, Cartoon, and Sketch. Notably, the Sketch domain presents a significantly different data profile compared to the other three. This disparity allows us to simulate a scenario of rapid distribution change, particularly noticeable during the transition to training on the Sketch domain. We observed that all tested methods, including ours, initially experience a performance drop in this phase. However, our method demonstrates a quicker recovery in accuracy than others, highlighting its adaptability to such abrupt changes in data distribution.
>
> We believe that rapid distribution changes pose a unique challenge: the new data can either be viewed as out-of-distribution (OOD), which may not be relevant and thus could be ignored, or as vital new information that should be incorporated into the model. This dichotomy is crucial in continual learning environments.
>
> Our approach specifically encourages the storage of data near the decision boundary and those instances within a class that exhibit the most significant differences. While this strategy is effective in capturing diverse data representations, it also raises the possibility of incorporating more OOD data into the memory set. This inclusion of OOD data could potentially interfere with the model's ongoing learning process, posing a challenge that we aim to address in future work.

---

### Official Review · Reviewer_ddu2 · 2023-11-02

**Soundness:** 3 good
**Presentation:** 2 fair
**Contribution:** 3 good
**Rating:** 3
**Confidence:** 4

**Summary:**

This paper proposes a storage sample selection strategy based on feature distance, which evaluates the distance between new data and the memory set to assess the representability of new data. And based on this, contrastive learning loss is introduced to leverages the importance weights computed during memory selection process. Experiments on different online incremental learning setting demonstrate the effectiveness of the proposed method.

**Strengths:**

1.	The proposed FDBS and IWL is of great importance and applicable for the different online incremental learning setting.
2.	Adequate and reasonable proof of derivation.
3.	Expensive experiments are conducted in different datasets.

**Weaknesses:**

1.	The overall paper is well written, however, some details need a little more attention. Such as the title is too long, a little difficult to understand the point of this paper.
2.	From the title and abstract of the paper, it looks like your method is supposed to work for all settings of online incremental learning, so it should achieve the best performance so far for different settings, but for experiments with balanced datasets, it looks like OnPro[1], GSA[2], and OCM[3] have achieved much better experimental results with the same resnet18 and M=5K methods, so you should add the latest methods into it to make your experiments more convincing

[1] Online Prototype Learning for Online Continual Learning, ICCV2023
[2] Dealing With Cross-Task Class Discrimination in Online Continual Learning, CVPR2023
[3] Online Continual Learning through Mutual Information Maximization, ICML2022

**Questions:**

Please refer to the strengths and weaknesses.

---

> ### Author Response · Authors · 2023-11-21
>
> Thank you for your valuable suggestion. We fully understand the importance of including the latest methods in our experiments to enhance the credibility and relevance of our work. However, we would like to highlight the differences in terms of the positioning and focus of our method compared to OnPro[1], GSA[2], and OCM[3], although these works represent important contributions to the field of online class incremental learning.
>
> 1. **Distinct Scenarios and Setting**: OnPro[1], GSA[2], and OCM[3] have brought significant contributions on continual learning but they are primarily proposed under class incremental learning scenario and tested on the Split CIFAR100 and Split mini-Imagenet  under data balanced setting. For instance, in section 3.1 on problem definition, OnPro[1] assumes that different tasks D_t are mutually disjoint in terms of classes. Therefore,  their method assumes that learning data for a given class can only arrive once through a number of successive batches within a given task. Such an assumption moves away from the original goal of continual learning where the learned knowledge is supposed to be consolidated anytime, including already learned concepts.   In contrast, our paper does not make such an assumption and proposes a novel online continual learning method operating under much more diverse and challenging scenarios, including Domain-incremental, Class-incremental, and Domain-Class incremental learning under both balanced and imbalanced settings. This broader scope addresses a wider range of practical applications and challenges in the field.
> 2. **Methodological Differences**: our method primarily concentrates on the efficient management and utilization of the memory buffer which is the cornerstone of all rehearsal-based CL methods. This approach is  fundamentally orthogonal to the methods mentioned above, emphasizing different aspects of the learning process. Therefore, it is more appropriate for our method to be compared with other memory selection methods, which align more closely with our research's core focus:
>     - OnPro[1] does a great work in feature learning to avoid shortcut learning, using in particular the contrastive learning loss on prototypes from the memory set and the current batch and data augmentation. It employs Experience Replay (ER) to select memory samples. In contrats, our method introduces a new, more efficient memory selection technique compared to ER. Their method does a great work on calculating the contrastive prototype learning loss within the memory set and within current batch. In contrast, our method generates the contrastive learning loss between the memory set and current batch without data augmentation.
>     - GSA[2] introduces a new training objective with a gradient-based self-adaptive loss to tackle gradient imbalance (GI).
>     - OCM[3] maximizes mutual information between input data and holistic feature representations, using local and global rotations for data augmentation.
>
> Experimentally, we have shown in our paper that our memory selection method (FDBS) can be combined with any other rehearsal-based CL methods to obtain improved performance and robustness to unbalanced data. In the following section, we further showcase this property of complementarity of our proposed memory selection method when it is combined with OnPro[1].
>
> 3. **Comparative Analysis with OnPro[1]**:
>     - In balanced Class-incremental Learning (CIL) on CIFAR100, OnPro shows superior performance with a 35.98% accuracy compared to our 21.13%. This result suggests the effectiveness of feature learning through online prototype learning in OnPro[1] under data balanced setting;
>     - In the more complex and realistic scenario of imbalanced Domain-incremental Learning (DIL) on the PACS dataset, our  proposed method (FDBS+IWL) outperforms OnPro, achieving 31.25% accuracy against their 26.54%.
>     - If we replace OnPro's memory selection method with our FDBS, we observe that in a balanced scenario, the resultant method increases slightly the performance. However, in the context of imbalanced domain-incremental learning, combining OnPro with our method achieves the best performance, further demonstrating the effectiveness and the versatility of our memory selection approach.
>
> These results along with those in the submission across different scenarios suggest that OnPro and similar methods are highly specialized for balanced CIL. In contrast, our proposed method, focusing on efficient memory management, demonstrates greater versatility and effectiveness in these diverse and more challenging learning environments, and its ability to be combined with any rehearsal-based CL method.
>
>
> |       | Balanced CIL (Cifar100) | Imbalanced DIL (PACS) |
> |-------|-------------------------|-----------------------|
> | OnPro | 35.98                  | 26.54                 |
> | Ours  | 21.13                   | 31.25                 |
> |OnPro + FDBS(ours)| 36.2|  33.14|

---

### Meta-Review · Area_Chair_qbTm · 2023-12-11

**Metareview:**

In this paper, the authors propose a method for online continual learning. The reviewers keep their negative ratings after the rebuttal period. After checking the paper, I think the quality of this paper is not good enough to be presented at a top-tier conference. Many major concerns are not addressed, e.g., the baseline methods used for comparison are outdated, with the most recent one dating back to 2022.

Therefore, the final decision is reject.

**Justification For Why Not Higher Score:**

As I mentioned in my meta-review, many major concerns are not addressed, e.g., the baseline methods used for comparison are outdated, with the most recent one dating back to 2022. The authors gave their responses during the rebuttal period. However, I don't think the responses are convincing enough to overturn the reviewers' decisions.

**Justification For Why Not Lower Score:**

N/A

---

### Decision · Program_Chairs · 2024-01-16

Reject